# Students’ Burnout at University: The Role of Gender and Worker Status

**DOI:** 10.3390/ijerph191811341

**Published:** 2022-09-09

**Authors:** Caterina Fiorilli, Daniela Barni, Claudia Russo, Vanessa Marchetti, Giacomo Angelini, Luciano Romano

**Affiliations:** 1Department of Human Sciences, University of Rome, LUMSA, 00193 Rome, Italy; 2Department of Human and Social Sciences, University of Bergamo, 24129 Bergamo, Italy; 3Department of Human Sciences, European University of Rome, 00163 Rome, Italy

**Keywords:** burnout, well-being, university students, gender, working students

## Abstract

Students’ burnout has been widely investigated in recent decades, mainly showing a higher risk for female students across academic levels. To our knowledge, few studies have investigated whether employed students experience higher academic burnout risks. In this regard, previous findings have shown mixed results. The current study investigated the differences in burnout experience based on students’ gender and worker status. We expected to find differences among study groups in their burnout levels. The participants were 494 Italian university students (49.6% female students; 49.4% working students) who completed the short version of the Burnout Assessment Tool Core dimensions (BAT-C). Firstly, we investigated the BAT-C measurement invariance across gender and worker status subgroups. Secondly, a multivariate analysis of variance (MANOVA) showed significant gender differences in burnout levels. Specifically, female students showed higher levels of exhaustion, cognitive impairment, and emotional impairment than male students. Nevertheless, no interactive effects between gender and worker status were observed in the current sample. To sum up, gender is a key factor for understanding several BAT symptoms, and it should be considered by academic staff interested in preventing burnout at university and its dropout consequences.

## 1. Introduction

Student burnout is a health issue that has been widely investigated in recent decades. It is a study-related syndrome characterized by a maladaptive emotional and physiological response to long-term exposure to stressful events affecting high school students [1,2,3,4,5,6] and university students (e.g., [7]). Schaufeli et al. [7] stated that “burnout among students refers to feeling exhausted because of study demands, having a cynical and detached attitude toward one’s study, and feeling incompetent as a student” (p. 465). Overall, burnout occurs when students feel overwhelmed and exhausted without having (or feeling they do not have) the effective resources to face prolonged stressful events (e.g., [8,9,10]). Salmela-Aro et al. [11] applied the demands–resources model to students’ experiences, assuming that “the more study-related demands, such as pressure and workload, the students experience, the more study burnout they experience” (p. 2). Like work, study implies several mandatory and pressing tasks, such as homework, deadlines for handing in assignments set by the teacher, and preparation for intercourse tests or tests at the end of the academic year, with many negative outcomes for students.

Most studies on students’ burnout have involved primarily medical students, reducing the generalizability of the observed phenomenon. This is probably because of many distressing activities during medical school (e.g., attending courses and practical experience in the hospital environment) [12]. According to the systematic review and meta-analysis by Erschens et al. [13], the percentage of burned-out medical students ranges from 7% to 75.2%. Other studies [12,14] have noted that at least half of medical students have experienced burnout during their academic courses. Finally, a more recent systematic review [15] has shown that university students with severe burnout symptoms (particularly in healthcare courses) varied from 30.5% in high-income countries to 54.5% in low- and middle-income countries. Yet, Asikainen et al. [16] have found that a quarter of their sample of university students attending life sciences courses at a large Finnish university experienced burnout symptoms. However, empirical data on university students’ burnout have not been systematically collected across countries using the same methods and instruments, making cross-cultural comparisons difficult.

Various instruments in the literature are used to assess students’ burnout (for a review, see Madigan and Curran, [17]), generally focused on three dimensions of the burnout construct, namely emotional exhaustion (i.e., feeling of tiredness and fatigue), cynicism (i.e., feeling of distance from school-related activities), and inadequacy (i.e., feeling incompetent as a student). These dimensions, originally assessed concerning work-related burnout, were then adapted to investigate study-related burnout (e.g., [17,18]). According to Madigan and Curran [17], while exhaustion is the core of academic burnout, cynicism and inadequacy are considered the consequent behavioral and emotional expressions of exhaustion. The major risk of these instruments is oversimplifying the multifaced nature of burnout, which is characterized by several primary symptoms. Schaufeli et al. [19] have recently validated a new tool to assess a plethora of symptoms referring to burnout syndrome: the Burnout Assessment Tool (BAT). The first four dimensions of the BAT (i.e., BAT core version, BAT-C) comprise the core symptoms of burnout, i.e., exhaustion (i.e., fatigue and tiredness), mental distance (i.e., detachment), cognitive impairment (i.e., attentional and memory difficulty), and emotional impairment (i.e., emotion regulation management difficulty). The BAT, also available in a short version, is inspired by a more comprehensive approach that capitalizes on the long-tradition research where strict relationships among emotional, cognitive, and behavioral dimensions in burnout syndrome have been confirmed [19]. In the last two years, it has been translated and adapted to be used in several countries for adult workers (e.g., [20,21,22,23]), high-school students [24], and recently university students [25], showing brilliant psychometric properties.

Research has consistently shown negative associations between students’ burnout and academic performance, engagement, self-efficacy, self-esteem, and positive coping strategies (see, for example, [26,27,28]). Similarly, academic burnout is positively associated with detachment and cynicism in the university context [26,29,30] and negatively with academic achievement [17]. Moreover, several studies have shown that university students with burnout symptoms tend to show a high risk of developing eating disorders [31], sleep disorders [32], addiction [33], and mental health issues [34]. Consequently, students’ burnout can strongly threaten their careers and the opportunity to conclude their academic paths positively [24,35,36,37].

However, students’ reactions to stressful events are highly diverse, and the burnout experience is extremely subjective. In the current study, we focused on two socio-demographic characteristics, namely students’ gender and worker status, expected to impact students’ burnout levels.

Gender is a key dimension when burnout is analyzed from students’ perspectives. Findings from adolescent students have demonstrated that girls across education levels feel more exhausted compared to their male classmates [24,38,39]. However, while female students are more at risk of experiencing exhaustion and inadequacy compared to their male classmates, no gender differences have been observed, considering cynicism [40].

In studies involving university students, findings are not consistent regarding students’ gender effects. For example, in some studies among medical students, women were more at risk (e.g., [41]), while in others, this result was not confirmed [42,43]. Furthermore, few studies have shown the highest burnout level in male students [44,45]. Recently, Asikainen et al. [16] conducted a latent profile analysis showing that while female university students were overrepresented in the high exhaustion and inadequacy profile, no gender differences in cynicism emerged, which is consistent with previous research results on academic burnout (e.g., [40]). Taken together, these studies seem to suggest that gender differences in academic burnout levels still require a deep investigation to understand whether and to what extent female students are more vulnerable to burnout than male students. With this in mind, it is interesting to investigate the gender effects using a wider approach to burnout symptoms, as proposed by Schaufeli [19].

A second key dimension describing university students, which could be related to their burnout experience, is the distinction between working students (WSs) and non-working students (NWSs). Earlier research has shown that working students were more stressed and anxious than those colleagues who were not employed while attending the courses (e.g., [46]). More recent studies have pointed out that, due to a dual responsibility request, students’ employment is an additional stressful source threatening their mental health and academic achievements (for a review, see Galbraith and Merrill, [47]). Balancing academic and job demands can lead to greater personal challenges and, therefore, an increase in frustration, stress, and burnout [48]. An interesting work by Draghici and Cazan [25] analyzed the burnout levels in a sample of employed university students by comparing work-related and academic-related experiences. Findings have shown that academic burnout is higher than work-related burnout. However, on the other side, some findings provide a different picture. For example, Zimitat [49] and Dundes and Marx [50] found that being a WS leads to better long-term performance, while Mounsey et al. [51] found no differences in academic achievements between the two samples. These inconsistent results have been explained according to two different perspectives. First, the unexpected positive picture observed in the WS sample may have been a result of their multiple and simultaneous roles and responsibilities. The experience of the WS is challenging and can impact academic and personal outcomes [52], but it may also promote high self-awareness skills and good strategies to manage risks and resources (e.g., [53,54]). Second, the expected negative condition of the WS emerging from other studies could be due to the multiple stressful sources co-occurring in the WS’s life. For example, loading tasks, time pressure, and reduced time spent on social events are likely associated with low mental health (e.g., [55]).

While psychological studies have explored possible gender differences in academic (e.g., [39]) and work-related burnout (e.g., [56]), no studies have investigated these differences in depth by combining academic and work experiences. In other words, little is known about possible gender differences in WSs regarding academic burnout. Several studies have shown that women are subject to greater work pressures, difficulties obtaining career advancement, lower pay, lack of support, and more harassment than their male colleagues [57,58,59]. All the aspects could contribute significantly to increasing sources of stress for female workers and contribute to an enhanced risk of burnout [60]. Thus, it is interesting to investigate burnout experiences by comparing WSs and NWSs and considering possible gender differences.

The literature mentioned before clearly shows a gap in research concerning the differences in burnout experience based on students’ gender and worker status, which have not yet been sufficiently studied. Investigating these differences instead becomes cogent, given the increasing spread of burnout among university students. Concerning the gender effect, we expected to find differences between female and male students’ burnout levels (H1). Furthermore, in light of the conflicting results in the literature, we explored burnout differences between working and non-working students. More specifically, we expected that employed students would be the most at risk of experiencing burnout (H2). Aligning H1 and H2, we further analyzed the possible mean differences in burnout dimensions across the interaction between gender and worker status. Before testing these hypotheses, we analyzed the BAT-C short version invariance across gender and worker status to ensure that measurements in the different groups were comparable.

## 2. Materials and Methods

### 2.1. Participants

The sample consisted of 494 Italian university students (female = 49.6%; WS = 49.4%), with ages ranging from 20 to 35 years (M = 25.64, SD = 4.75). Students enrolled in two main academic curricula (i.e., humanistic vs. scientific curriculum). Furthermore, students were enrolled in different years of the degree (i.e., from first year to out of regular course). Finally, the sample comprised working and non-working students (for details, see Results section).

### 2.2. Procedure

Participants were recruited by a snowball sampling method using social media and were asked to complete an anonymous online questionnaire. Only volunteer students who gave informed written consent were involved in the study, and all the study procedures were conducted following the Declaration of Helsinki of 1964 and its latest version (2013). The study received ethical approval from the Ethics Committee of the LUMSA University of Rome, Italy.

### 2.3. Instruments

The Italian version of the Burnout Assessment Tool–Core Symptoms short version (BAT-C-Short version; [19,20,24,60]) (see Appendix A) was used to assess burnout. It is composed of 12 items and four dimensions: exhaustion (3 items: e.g., “I feel mentally exhausted”), mental distance (3 items: e.g., “I struggle to find enthusiasm for studying”), cognitive impairment (3 items: e.g., “I have trouble staying focused”), and emotional impairment (3 items: e.g., “I feel unable to control my emotions”). All items were rated on a five-point Likert scale (1 = “never”, 5 = “always”).

The questionnaire asked for sociodemographic information (i.e., age, gender, and worker status) as well as information concerning academic curricula and year of enrollment. Employment status was determined by a single item asking whether the student was regularly employed in the previous academic year (i.e., with yes or no as possible responses).

### 2.4. Data Analysis

Preliminarily, since the BAT-C short version for university students has not been previously validated within Italian samples, we first tested whether it could be a valid measure of burnout in the groups we considered (i.e., male university students vs. female university students, WSs vs. NWSs). Specifically, we tested the factorial structure and the measurement invariance across the groups. When measurement invariance is confirmed, comparing and interpreting group means may be carried out meaningfully [61]. Measurement invariance requires verifying that the factorial structure (i.e., configural invariance), items’ factor loadings (i.e., weak invariance), items’ intercepts (i.e., strong invariance), and items’ residual variances (i.e., strict invariance) are equivalent across the compared groups (e.g., WS and NWS) [62]. To assess whether the strong measurement invariance was achieved, we ran a multistep procedure through which each step was compared with the less constrained model (e.g., weak invariance vs. configural invariance) [63]. When the sample size is larger than 300, as in our case, the delta Comparative Fit Indices test (ΔCFI) is recommended rather than the delta chi-square difference test (Δχ^2^) [64]. A ΔCFI not significantly more than 0.01 across the models indicates that the model fit does not deteriorate considerably when further constraints are added. Moreover, Chen [65] suggested considering the delta Root-Mean-Squared Error of Approximation (ΔRMSEA) ≤ 0.015 to assess scalar invariance. Moreover, besides the ΔCFI and the ΔRMSEA criteria, each model should show a good data fit for the model to be considered acceptable. We considered the model fit as an acceptable fit of data when the CFI value was 0.90 or higher, and the Standardized Root Mean Square Residual (SRMR) and RMSEA were 0.08 or lower (for more details regarding fit indices, see [65,66,67]). After assessing measurement invariance, we tested the scales’ internal consistency, separately for each dimension (i.e., McDonald’s omega). The measurement invariance across gender and students’ working condition were verified with Mplus v. 6.0 (Muthén & Muthén, Los Angeles, CA, USA) [68]. Internal consistency and MANOVA were verified using Jamovi v. 1.6 (The Jamovi project, Sydney, Australia) [69] and SPSS v. 23.1 (Statistical Product and Service Solutions, Chicago, IL, USA) [70].

Secondly, we performed an exploratory chi-square test (χ^2^) of association in order to verify if there were significant differences in the distribution of participants for gender and worker status concerning academic curricula and year of enrollment. Then, to test our hypotheses, a multivariate analysis of variance (MANOVA) 2 (gender: male students vs. female students) × 2 (students’ worker status: WS vs. NWS) was performed to assess mean differences in burnout dimensions across gender, worker status, and their interaction (i.e., gender*worker status). We used the MANOVA to account for the theoretical and empirical intercorrelations between the BAT-C dimensions [71]. To perform the analysis, the score of the BAT-C dimensions (i.e., exhaustion, mental distance, cognitive impairment, and emotional impairment) was calculated by averaging the items composing each dimension.

## 3. Results

### 3.1. Measurement Invariance of BAT-C Short Version

In Table 1 and Table 2, the results of the measurement invariance analysis are reported. Specifically, in the tables, there are the fit indices related to:The confirmatory factor analysis of the BAT-C conducted in the whole sample;The configural invariance model in which we verified whether the structure of BAT-C was the same across groups;The weak invariance model in which we constrained the item loadings on the factors to be equal across the groups;The strong invariance model in which we constrained the item intercepts to be equivalent across the groups;The strict invariance model in which we constrained the item residuals across groups.

Each step was compared with the less constrained model through the ΔRMSEA and ΔCFI.

Results indicate that the BAT-C short version was strictly invariant concerning students’ gender. Specifically, BAT-C measures the four dimensions equally for male and female students (Table 1).

As observed for the measurement invariance across students’ gender, the BAT-C short version can be considered a valid instrument to assess burnout symptoms. Despite worker status differences, our results highlight that the instrument worked similarly across worker status, thus making possible the mean comparisons (Table 2).

### 3.2. Descriptive Statistics, Internal Consistency, and Multivariate Analysis of Variance (MANOVA)

Table 3 reports the frequencies and percentages concerning curricula and the year of enrollment for the total sample, and the frequencies and percentages split by gender and worker status.

Our sample comprised 249 (50.4%) males and 245 (49.6%) females. Of the total sample, 244 (49.4%) were working students, and 250 (50.6%) were non-working students. Most of our sample attended the humanistic curriculum (78.5%), and the other 21.5% attended the scientific curriculum. Concerning the year of enrollment, 37.4% of the sample was composed of first-year students, 20.2% were second-year students, and 22.5% were third-year students. A few students, 4.7%, were enrolled in the fourth year. The remaining 15.2% of students were out of the prescribed time or missing data.

The chi-square (χ^2^) test of association showed no significant differences in the distribution concerning worker status and academic curricula (χ^2^ = 0.01, df = 1, *p* = 0.938), as well as for worker status and year of enrollment (χ^2^ = 5.08, df = 4, *p* = 0.279). Concerning gender, the chi-square (χ^2^) test showed significant differences in the distribution concerning gender and academic curricula (χ^2^ = 5.37, df = 1, *p* = 0.020), while no significant differences in the distribution were observed concerning gender and year of enrollment (χ^2^ = 7.53, df = 4, *p* = 0.110).

Table 4 shows the descriptive statistics regarding mean and standard deviation and the MANOVA results.

The exhaustion, cognitive impairment, and emotional impairment dimensions of the BAT-C showed an excellent internal consistency (ω > 0.80), while an acceptable internal consistency was found for mental distance (ω = 0.65).

Consistent with the first hypothesis, the MANOVA results show significant gender differences (Wilks’ Lamba = 0.847, *p* < 0.01). In detail, female students had higher levels of exhaustion (M = 3.60; SD = 0.89) compared to male students (M = 2.97; SD = 0.89). Moreover, female students showed higher levels of cognitive impairment (M = 3.29; SD = 0.91) than male students (M = 3.04; SD = 0.98). Similarly, female students reported higher levels of emotional impairment (M = 2.73; SD = 1.02) than males (M = 2.27; SD = 1.00). No significant differences emerged considering the mental distance dimension (M_male_ = 2.89; SD = 0.87 vs. M_female_ = 2.88; SD = 0.81). Otherwise, it is interesting to note that the largest gender difference was found regarding exhaustion (ƞ^2^_p_ = 0.11). Therefore, these results highlight that, except for the mental distance dimension, females had significantly higher burnout levels than males when measured with BAT-C. Neither the worker status (Wilks’ Lamba = 0.992, *p* = 0.435) nor the interaction between gender and worker status (Wilks’ Lamba = 0.992, *p* = 0.445) was statistically significant.

Considering the slight but significant association between gender and academic curriculum (the Variable Inflation Factor (VIF) tested for gender and academic curricula concerning the four burnout dimensions was <0.05, thus revealing no multicollinearity among the variables), we carried out a multivariate analysis of covariance (MANCOVA)—with gender and worker status as independent variables and academic curriculum as the covariate—to corroborate the MANOVA results and control for possible confounding effects of academic curriculum. The MANCOVA results show no significant effects of the covariate (Wilks’ Lamba = 1.000, *p* = 0.997). The results concerning the effects of gender (Wilks’ Lamba = 0.848, *p* < 0.001), worker status (Wilks’ Lamba = 0.992, *p* = 0.435), and gender*worker status (Wilks’ Lamba = 0.992, *p* = 0.446) on the burnout dimension mean levels were unchanged. Specifically, only the main effect of gender was statistically significant, with female students reporting higher levels of exhaustion, cognitive impairment, and emotional impairment. On the contrary, due to the absence of significant associations among students’ year of enrollment and study variables, it was not further controlled as a covariate in the MANCOVA.

## 4. Discussion

In the current study, we were interested in investigating the role of students’ gender and worker status in their academic burnout experience. While previous research has extensively explored the gender effect on academic burnout (e.g., [72,73,74]), students’ condition as workers has been less investigated, particularly considering the interaction effect between gender and worker status. Furthermore, the existing findings which come from the classic measures of burnout (e.g., Oldenburg Burnout Inventory, OLBI [75,76]; Maslach Burnout Inventory–Student Survey, MBI-SS; School Burnout Inventory, SBI [77,78]) have neglected the variety of primary symptoms of burnout syndrome. With this in mind, we verified whether the BAT-C short version—used in our study to measure university students’ burnout as a complex and multifaced construct—remains valid across gender and worker status. Findings from our preliminary analyses point us in that direction. The BAT-C short version worked similarly among male and female students and working and non-working student groups, according to Zumbo and Koh’s [79] invariance test indices. Furthermore, the results concerning the measurement invariance across gender align with those recently reported by Sinval et al. [80] in their validation study of the BAT-C short version in a sample of Portuguese and Brazilian workers. Nonetheless, the current study is the first one to test the measurement invariance of the BAT-C short version across gender in a sample of university students.

Based on our first hypothesis, we expected significant differences in the levels of burnout between female and male students. Previous studies have found that burnout levels differ due to gender effects; however, results remain inconsistent, e.g., [41,42,43]. Our findings show higher levels of exhaustion, emotional impairment, and cognitive impairment in female students than in male students, even after controlling for academic curricula, while no differences were observed in BAT’s mental distance dimension. Although female students showed higher burnout symptoms than male students, their curriculum choice did not add new explanations for their maladaptive experience. Concerning our second hypothesis, no differences in the levels of burnout were found according to worker status or the interaction between students’ gender and worker status. Thus, in light of these findings, we can highlight the pivotal role of students’ gender when worker status was also considered. In the following paragraphs, we provide details and suggest further investigations to overcome the current study’s limits.

In the current study, female students were more at risk than male students to experience burnout in terms of exhaustion and cognitive and emotional impairment. This result is in line with the previous research described above, which referred to high school students (e.g., [24,38,39]) and, in a few cases, to university ones (e.g., [25,41]). Furthermore, the higher risk for females, particularly considering the exhaustion dimension, is a consistent result in most research addressing workers’ burnout (e.g., [56]), showing the strict relationship (e.g., for a review, see Koutsimani et al. [81]) and, though somewhat debated, the overlap [82] between depression and burnout constructs. Blanco et al. [83] noted that female university students tend to show a higher prevalence of depression than the general population. Effectively, it has long been demonstrated that the higher occurrence of depression in women than in men is due to several biological, sociological, economic, and political factors (e.g., [84]).

In our findings, female university students also reported higher cognitive and emotional impairment levels than male colleagues, as revealed by BAT’s measures. Of course, this evidence is not surprising since cognitive impairment is part of the same instrument. Nevertheless, this is particularly alarming considering that memory and attention impairments can last more than the other burnout symptoms [85], thus compromising academic achievement in the long term [86]. In their systematic review, Deligkaris et al. [87] found a strong association between burnout and cognitive function decline (i.e., executive functions, attention, and memory performances). This was also recently confirmed by Bianchi and Schonfeld [88], who found a relevant predictive role of employed adults’ depressive symptoms on cognitive performance decrease.

Additionally, prior findings have shown high emotional impairment risks for women, which means difficulties in managing negative emotions (e.g., irritability and reactivity) [89,90]. Nonetheless, this result could also be seen in light of previous studies suggesting that men are more reluctant to communicate their emotions to others than women (e.g., [91]). Further studies speculate that these differences are related to the socialization hypothesis, which postulates that traditional socialization patterns (e.g., the traditional female role is expected to be more emotionally expressive than the traditional male one) may affect people’s response to stress. In this regard, men avoid expressing feelings as it could demonstrate weakness [90,91].

These aspects might have contributed to a possible bias in the rating of the emotional impairment dimension and the observed mean differences in scores.

No gender differences emerged with regard to mental distance. According to some scholars [84,92], mental distance is a feeling of psychological detachment characterized by a scarcity of motivation, interest, and enthusiasm [84,92]. High levels of mental distance reveal a strong aversion toward the job and are related to feelings of cynicism toward one’s work [93,94]. While it was expected that all BAT-C dimensions were homogeneously expressed in female students, showing higher levels than in males, previous research (for a review, see [95]) also suggested other perspectives that could be considered. Burnout symptoms may also occur and develop heterogeneously across individuals. In this regard, it would be interesting to shift from a variable-centered approach to a person-centered approach that could better account for the subjective nature and heterogeneity of burnout experience.

Furthermore, contrary to our expectation, no differences were observed between WSs and NWSs in burnout dimension mean levels, and no interactive effects between gender and worker status were observed in the current sample. These findings are in contrast with some previous studies. According to some scholars (e.g., [96,97]), it may depend on the need to balance their multiple roles, known as the inter-role conflict. It is an interesting perspective supporting the high risk for employed students, even though it has not been confirmed in the current study or in earlier findings, e.g., [56].

Finally, as a part of the gender and worker status dimensions investigated here, students’ personal and external resources could be analyzed to cluster students and investigate their role in academic and work-related stressful events. Effectively, individuals may cope differently with them through several strategies and external support.

## 5. Limitations and Future Research Directions

The current study presents some important limitations to be considered in interpreting the results and in future research developments. First, we used a one-measure method to investigate students’ burnout levels. Even though the BAT-C evaluates a wide spectrum of symptoms, further similar stress indices, depression levels, and personal resources should be assessed to deeply understand students’ academic adjustment. Second, our study did not investigate WSs’ work-related variables (e.g., students’ working hours per week and kind of work), which might influence university students’ burnout experience. Even though no previous studies have investigated the effect of working conditions on university students’ burnout, the abundant literature on burnout at work has already demonstrated how several work-related characteristics impact workers’ well-being. In this regard, future research on working university students should collect and consider work-related variables’ effect on burnout levels. Third, the interaction between private and working life dimensions should be considered to understand, for example, the role of social support in coping with burnout experience. Fourth, cross-sectional studies neglect the effect of academic paths on students’ burnout [98]. Since academic burnout mainly involves adolescents and young adults, adopting a longitudinal approach to catch burnout development could be useful. For example, female and male students have shown different paths of burnout symptoms from high school to academic tracks [99]. Finally, the internal consistency of the mental distance dimension, measured by McDonald’s omega, was barely acceptable. According to a recent study by Kalkbrenner [100], to be acceptable, McDonald’s omega scores should be between 0.65 and 0.80 (and >0.80 to be strong). Moreover, previous studies considered scores between 0.60 and 0.70 as questionable but acceptable (e.g., [20]). Interestingly, a similar result concerning the mental distance dimension (e.g., a lower score than the other BAT dimensions when testing for internal consistency) has already been observed in a previous adaptation/validation of the BAT with Italian samples [20,24], thus suggesting that further investigations should be conducted concerning this BAT dimension.

## 6. Conclusions

The BAT-C short version is overall a good self-report instrument which overcomes the limits of the traditional burnout evaluation based on the three classic factors. More specifically, this tool is usable even with different student populations, providing a wider spectrum of symptoms and offering a valid instrument for further cross-culture and comparative investigations. Importantly, our findings show that gender is a significant dimension for an initial understanding of university students’ burnout experience. Two practical implications can be drawn from these findings in order to prevent burnout. First, female students are more vulnerable to burnout experience and especially to feeling drained and emotionally exhausted, unable to cope, tired, and down. Second, the greater attention paid to women’s high risk could neglect men’s experience of burnout. Actually, in terms of risk factor evaluation, female students have high emotional exhaustion levels on the one hand, but, as shown by previous studies, they are also more engaged in their academic careers on the other one. On the contrary, male students show a low level of exhaustion but are highly cynical toward the academic environment [101]. All these dimensions might affect women’s and men’s academic careers differently and therefore require different prevention programs.

## Figures and Tables

**Table 1 ijerph-19-11341-t001:** Measurement invariance between male and female groups.

Invariance	χ^2^	Df	RMSEA	CFI	SRMR	ΔRMSEA	ΔCFI
Total sample	103.571	48	0.048	0.981	0.032	-	-
Configural	177.502	96	0.059	0.971	0.040	-	-
Weak	195.985	104	0.060	0.967	0.050	0.001	−0.004
Strong	216.495	112	0.061	0.963	0.054	0.001	−0.004
Strict	210.702	124	0.060	0.961	0.057	−0.001	−0.002

**Table 2 ijerph-19-11341-t002:** Measurement invariance between WSs and NWSs.

Invariance	χ^2^	Df	RMSEA	CFI	SRMR	ΔRMSEA	ΔCFI
Total sample	103.571	48	0.048	0.981	0.032	-	-
Configural	172.397	96	0.057	0.974	0.043	-	-
Weak	177.439	104	0.053	0.975	0.046	−0.004	0.001
Strong	194.809	112	0.055	0.972	0.048	0.002	−0.003
Strict	210.702	124	0.053	0.970	0.051	−0.002	−0.002

**Table 3 ijerph-19-11341-t003:** Descriptive statistics (frequencies and percentages) of the total sample and split by gender and worker status for the academic curricula and year of enrollment.

Descriptives	Total Sample (N = 494)	Gender		χ^2^ Test for Gender	Worker Status		χ^2^ Test for Worker Status
	N (%)	Male N (%)	Female N (%)		WS N (%)	NWS N (%)	
		249 (50.4%)	245 (49.6%)		244 (49.4%)	250 (50.6%)	
*Academic curricula*				*p* = 0.020			*p* = 0.938
Humanistic curriculum	388 (78.5%)	185 (74.3%)	203 (82.9%)		192 (78.7%)	196 (78.4%)	
Scientific curriculum	106 (21.5%)	64 (25.7%)	42 (17.1%)		52 (21.3%)	54 (21.6%)	
*Years of enrollment*				*p* = 0.110			*p* = 0.279
First year	185 (37.4%)	101 (40.6%)	84 (34.3%)		84 (34.4%)	101 (40.4%)	
Second year	100 (20.2%)	50 (20.1%)	50 (20.4%)		48 (19.7%)	52 (20.8%)	
Third year	111 (22.5%)	43 (17.3%)	68 (27.8%)		55 (22.5%)	56 (22.4%)	
Fourth year	23 (4.7%)	13 (5.2%)	10 (4.1%)		16 (6.6%)	7 (2.8%)	
Out of course	50 (10.1%)	25 (10.0%)	25 (10.2%)		26 (10.7%)	24 (9.6%)	
Not specified	25 (5.1%)	17 (6.8%)	8 (3.3%)		15 (6.1%)	10 (4.0%)	

Note. WS = working student; NWS = non-working student; χ^2^ = chi-square test of association.

**Table 4 ijerph-19-11341-t004:** Means, standard deviations, and MANOVA results for gender and worker status.

			M (SD)								MANOVA					
											Gender		Worker Status		Gender × Worker Status	
	ω	Total (N = 494)	Male Students (N = 249)	Female Students (N = 245)	WS (N = 244)	NWS (N = 250)	Male WS (N = 129)	Male NWS (N = 120)	Female WS (N = 115)	Female NWS (N = 130)	F(1,493)	ƞ^2^_p_	F(1,493)	ƞ^2^_p_	F(1,493)	ƞ^2^_p_
EXH	0.85	3.29 (0.95)	2.97 (0.89)	3.60 (0.89)	3.22 (0.95)	3.35 (0.94)	2.98 (0.91)	2.97 (0.89)	3.50 (0.92)	3.69 (0.86)	59.789 **	0.11	1.340	0.00	1.473	0.00
MD	0.65	2.89 (0.84)	2.89 (0.87)	2.88 (0.81)	2.84 (0.87)	2.94 (0.81)	2.85 (0.91)	2.93 (0.82)	2.81 (0.83)	2.94 (0.79)	0.033	0.00	1.722	0.00	0.102	0.00
CIMP	0.85	3.16 (0.95)	3.04 (0.98)	3.29 (0.91)	3.09 (0.95)	3.24 (0.95)	3.02 (1.00)	3.06 (0.98)	3.16 (0.91)	3.40 (0.90)	7.937 **	0.02	2.659	0.00	1.268	0.00
EIMP	0.82	2.50 (1.03)	2.27 (1.00)	2.73 (1.02)	2.41 (1.03)	2.58 (1.03)	2.27 (1.00)	2.27 (1.00)	2.56 (1.05)	2.87 (0.97)	24.062 **	0.05	2.922	0.01	2.986	0.01

Note. ƞ^2^_p_ = partial eta squared; ** *p* < 0.01; EXH: exhaustion, MD: mental distance, CIMP: cognitive impairment, EIMP: emotional impairment; ω = McDonald’s omega; McDonald’s omegas were calculated for the whole sample.

## Data Availability

All data generated or analyzed during this study are included in this article. Further enquiries can be directed to the corresponding author.

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
