# Peer review of "Students’ Burnout at University: The Role of Gender and Worker Status"

_ijerph, 2022, doi:10.3390/ijerph191811341_

Round 1

Reviewer 1 Report

This study concerning the possible relationships between university students’ gender and employment is scientifically well performed and clearly described. The study  is based on the instrument BAT-C and an extensive evaluation of the instrument is described. I only have some minor comments related to the content of the presentation in abstract and background and also in the methods.

Abstract: I suggest to include the full title of the BAT-C already in the abstract (I suggest  to exclude the full title for MANOVA in case of lack of space). In the background/method: you have collected information on what kind of studies the students perform. I would like to have seen a comment about why you did not take this into consideration (too few in each group?) since, as you rightly comment in the discussion: there is a well known overrepresentation of students in health care occupations presenting with burn-out and anxiety at Students’ health care units. Lastly, and maybe most importantly: in the methods section: how were the students recruited, what information did they get beforehand? Could the recruitment have caused a selection bias? 

Author Response

Reviewer 1.

This study concerning the possible relationships between university students’ gender and employment is scientifically well performed and clearly described. The study is based on the instrument BAT-C, and an extensive evaluation of the tool is described. I only have some minor comments related to the content of the presentation in the abstract and background and also in the methods.

Abstract: I suggest to include the full title of the BAT-C already in the abstract (I suggest to exclude the full title for MANOVA in case of lack of space).

Answer: Thank you for the suggestions. We have done it.

In the background/method: you have collected information on what kind of studies the students perform. I would like to have seen a comment about why you did not take this into consideration (too few in each group?) since, as you rightly comment in the discussion: there is a well known overrepresentation of students in health care occupations presenting with burn-out and anxiety at Students’ health care units.

Answer: Thank you. Yes, We agree with you. It is a caveat that the current study didn’t consider other confounding variables. Based on your valuable recommendation, we have controlled for the “type of study” in order to exclude a possible confounding effect of this variable on the levels of burnout. For this, we performed a MANCOVA 2x2 (gender*working status) in which the type of study (scientific curriculum vs human science curriculum) was used as a covariate. The MANCOVA results completely confirmed those of the previous MANOVA, i.e., a significant effect of gender, with female students reporting higher levels of exhaustion, cognitive impairment, and emotional impairment, but no significant effects of worker status, gender*worker status, or type of study. Given this result, we have decided not to include the new analysis in the manuscript, whose theoretical framework is focused on gender and working status.

Please see lines 127-133

Lastly, and maybe most importantly: in the methods section: how were the students recruited, and what information did they get beforehand? Could the recruitment have caused a selection bias? 

Answer: Thank you for your recommendations. We collected a convenience sample using a snowball method. We specified that the sample was closely balanced regarding gender and working status. Please see lines 195-196 and 210-212.

Reviewer 2 Report

Thank you for giving me the opportunity to read and review this interesting manuscript. It covers a highly relevant topic and examines research questions that are of interest for people attending university but also for employees workng at universities, who might have the power to form a study environment that helps to prevent student burnout. My specific comments can be found below.

Abstract

Line 21: measurement invariance

The remaining comments about the abstract correspond to the comments in the main sections of the manuscript.

Introduction

Line 32: It is recommended to include some references for the first sentence.

I think it would be important to make clear right from the start whether these considerations refer to students in general, i.e. students attending school, and also students attending university etc., or specifically to university students.

Lines 48 to 60: It would be nice to read something about potential reasons for those remarkable differences in student burnout prevalences between countries and studies.

Line 66: These dimensions were not "developed". Instead, the instruments trying to assess these dimensions were developed.

Line 95: You might consider writing "are highly diverse".

Lines 104 and 105: feel more exhausted ... compared to their male classmates, ...

Line 108: This has to be more exact. What do you mean by "at risk"? Like this, it could be understood as if every woman was at risk for facing burnout just by being female.

Lines 114 to 117: You introduce your thoughts on university students with "in studies that addressed university students, findings are more inconsistent regarding students' gender". Then, you consequentially back it up with study findings that underline the homogeneity regarding potential gender effects. However, in the end you assume that female university students are more likely to face specific burnout symptoms than male students. In my opinion, you are not consistent here.

Line 149: ... little (practically nothing) ... --> Please decide for one of the two here.

Line 150: WS instead of student workers

Lines 150 to 152: Language. Please consider to rephrase the sentence.

Line 154: more harassment

Lines 162 to 164: Please see the comment above. I do not agree that the literature you refer to really indicated that female university students are assumed to face a greater risk of burnout. If you want to present this assumption, you have to underline it more clearly with respective literature.

Materials and Methods

Lines 184 to 186: After reading these lines, I thought it would also be interesting to examine whether there is an effect of the number of semesters studied on burnout. Maybe you could include this in the analysis as a minor research question.

Line 188: Please describe the recruitment of the study sample in a bit more detail.

Line 209: When instead of whether

Lines 212 and 213: loadings, intercepts, variances

Lines 210 to 217: Please add an exlanatory sentence that, for example, for weak measurement invariance it is assumed that factor loadings of the model are equal in both groups.

Lines 219 to 221: Language, consider editing.

Lines 225 to 228: We considered model fit to be acceptable when ... 0.90 or higher, and... 0.08 or lower.

Line 232: students' working status: Please briefly define at some point how you differentiate worker students from non-worker students. Several aspects should be considered in my opinion.
How many hours of work per week are required to be considered as a worker student? Would one hour be enough?
How regularly should the student be engaged in working? Should he/she work every week on a regular basis? Or am I also considered as a WS when I was working in the last semester break three months ago? I think these cases should be clearly discussed, if not included in the analyses. In my opinion, regarding the acute demands and stress (at the time point when the measurements took place), it makes a difference if I have to work during the assessment week or if my last job was three months ago.

Lines 234 to 236: Language; you did not perform measurement invariance or internal consistency. You conducted some analyses to verify these criteria.

Line 235: I suggest using the term internal consistency, both here and throughout the manuscript.

Results

Lines 239 and 247: ...shows the fit indizes as well as ΔRMSEA and ΔCFI for...

Lines 239 to 241: Please again briefly describe the factor structure that you modeled.

Line 244: for instead of between

Lines 251 to 252: ...the BAT-C short version worked well in our sample. --> This is too vague and does not transport any specific information.

Lines 252 to 253: In detail, reaching the strict measurement invariance model, ... --> Language, please consider editing.

Table 3: Please indicate at an earlier point of the manuscript that you calculated sum scores for the subscales and not mean scores. Besides, I would recommend to use mean scores for the subscales since these are more intuitive knowing that you used 5-point Likert scales.

Table 3: You might want to discuss a potential bias in the ratings of emotional impairment since men might tend more to disguise their emotions towards others and also towards themselves or might feel weak when showing (or indicating) emotional problems.

Table 3: In view of hypothesis 2, I feel it would make sense to indicate the values on the burnout dimensions not only for female and male students and WS and NWS but also specifically for the subgroups stemming from the combination of these two factors.

Table 3: Please also indicate what ** and Æž2p mean.

Lines 264 to 265: This should be the first sentence after the table because an at least acceptable internal consistency is an important requirement for the analyses that are to follow. Additionally, I think that the omega of the mental distance subscale is not really good. This should be stated more clearly.

Lines 270 to 272: Language; consider editing.

Discussion

Lines 283 to 300: From line 283 to line 300 you elaborate on your analysis and results regarding the measurement invariance of the BAT-C. To me this feels like a little bit too much and I suggest to substantially shorten these statements in order to present a brief summary of what you have done, why you have done that and what you found out regarding measurement invariance. The way you have described it so far, it seems like a main result although it rather is the verification of a basic requirement for the following analyses.

Lines 298 to 300: Language

Lines 302 to 305: Please see the comments above concerning the justification of this hypothesis.

Line 313: In the section below, ... / In the following paragraph, ...

Line 315: In line with...

Line 316: ... are affected by students' gender ... --> This might be correct when speaking about it from a purely statistical point of view. However, you cannot say without a doubt that gender has a direct effect on the BAT dimensions. You may say that female and male university students differ in most of the dimensions.

Lines 328 and 329: ...higher cognitive and emotional impairment than male university students, ...?

Lines 332 to 338: I think the relationship between burnout and cognitive impairment/functioning should be put in perspective. The relationship is not really surprising since cognitive impairment represents one of the aspects with which burnout is assessed in this study.

Line 345: ... does not surprise us. --> I recommend to re-phrase this sentence since the reader might ask why you did not present a more specific hypothesis 1 given the fact that you already assumed that there are no differences in MD between females and males.

Lines 352 to 359: Here, I do not understand the point you want to make. I do not see the connection between the non-existent differences in MD between females and males and the suggestion of using a person-oriented approach. Please clarify.

Lines 360 to 366: I suggest shortening the first part of the paragraph to avoid repetitions.

Lines 368 and 369: no repetition needed

Lines 372 to 390: This paragraph contains valuable thoughts in my opinion. However, these thoughts do not relate closely enough to the present study and its findings, which makes it hard to follow. I have the feeling that the reader needs to find his/her own way of how to understand what is written in the text. Therefore, I think it would be helpful to make the connection of these considerations to what has been assessed and to what has been found out easier to understand.

I recommend to include a paragraph in the discussion section that clearly describes the strengths and limitations of the study.

It would be good to include a paragraph already in the discussion section that outlines some concrete recommendations regarding what could be done to prevent student burnout in the risk group of female university students. Please explain in more detail what could be deduced from the results.

Conclusions

Line 394: student populations

Lines 406 to 410: This is already written in the first paragraph of the section.

The conclusion section would benefit from two or three clear messages that describe what the study findings mean for the life of university students.

Author Response

Answer to Reviewer 2.

Dear Reviewer,

Thank you very much for your valuable revision. We agree with your concerns and are grateful for your useful suggestions and corrections. We hope you'll find the revised manuscript improved and worthy of publication.

In the following section, you'll find how we revised the manuscript step-by-step.

Sincerely,

The Authors

Abstract

Line 21: measurement invariance

Answer: thank you. We’ve corrected the expression. Please see line 22

The remaining comments about the abstract correspond to the comments in the main sections of the manuscript.

Introduction

Line 32: It is recommended to include some references for the first sentence.

I think it would be important to make clear right from the start whether these considerations refer to students in general, i.e. students attending school, and also students attending university etc., or specifically to university students.

Answer: thank you for your suggestions. According to the literature, we specified that the syndrome was observed in high-school and university students. Please see lines 35-37

Lines 48 to 60: It would be nice to read something about potential reasons for those remarkable differences in student burnout prevalences between countries and studies.

Answer: The main issue to that point is the different burnout measures used in each country, which makes any comparison and interpretation of that a hazard. Anyway, we think that a clearer expression might better explain the meaning. Please see the red lines in the text.

Line 66: These dimensions were not "developed". Instead, the instruments trying to assess these dimensions were developed.

Answer: Yes, you’re right. We corrected the expression.

Line 95: You might consider writing "are highly diverse".

Answer: We agree with you. We’ve accepted your suggestion. Thank you.

Lines 104 and 105: feel more exhausted ... compared to their male classmates, ...

Answer: Yes, we changed the expression. Thank you for the suggestion.

Line 108: This has to be more exact. What do you mean by "at risk"? Like this, it could be understood as if every woman was at risk for facing burnout just by being female.

Answer: ok. In this regard, we’ve canceled some lines and changed the sentences' general meaning, reducing the causal relationship between being a girl and, at the same time, being exhausted. Please see lines 109-116

Lines 114 to 117: You introduce your thoughts on university students with "in studies that addressed university students, findings are more inconsistent regarding students' gender". Then, you consequentially back it up with study findings that underline the homogeneity regarding potential gender effects. However, in the end you assume that female university students are more likely to face specific burnout symptoms than male students. In my opinion, you are not consistent here.

Answer: We agree with you and corrected the inconsistent argumentations regarding gender differences and our expectation to find that. Please see lines 127-133

Line 149: ... little (practically nothing) ... --> Please decide for one of the two here.

Answer: Yes. Done.

Line 150: WS instead of student workers

Answer: Yes. Done.

Lines 150 to 152: Language. Please consider to rephrase the sentence.

Answer: Thank you. We decided to cancel the sentence “Despite…workplace” which doesn’t give additional meaning to the general argumentation. Please see lines 169-170

Line 154: more harassment

Answer: Yes. Done.

Lines 162 to 164: Please see the comment above. I do not agree that the literature you refer to really indicated that female university students are assumed to face a greater risk of burnout. If you want to present this assumption, you have to underline it more clearly with respective literature.

Answer: Yes, we agree with you. In the past version, it was confounding at that point; now, we outlined the main results from the existing literature, which leads us to expect a high burnout risk for female students. Please see lines 182-183

Materials and Methods

Lines 184 to 186: After reading these lines, I thought it would also be interesting to examine whether there is an effect of the number of semesters studied on burnout. Maybe you could include this in the analysis as a minor research question.

Answer: Actually, we’ve considered this interesting hypothesis. However, the unbalanced between sub-groups prevents us from doing the measurement invariance control across this variable, which is an unavoidable starting point. Thus, we have emphasized that point in the limits and proposals section for further studies. 

Line 188: Please describe the recruitment of the study sample in a bit more detail.

Answer: Thank you for your recommendations. We collected data by a snowball sampling method. As we specified in the revised version of the manuscript.

Line 209: When instead of whether

Answer: Yes. Done.

Lines 212 and 213: loadings, intercepts, variances

Answer: Yes. Done.

Lines 210 to 217: Please add an exlanatory sentence that, for example, for weak measurement invariance it is assumed that factor loadings of the model are equal in both groups.

Answer: Thank you for your valuable suggestion. We have now added explanatory sentences regarding the measurement invariance steps and what they are aimed to assess. Please see lines 262-267.

Lines 219 to 221: Language. Consider editing.

Answer: Yes. Done.

Lines 225 to 228: We considered model fit to be acceptable when ... 0.90 or higher, and... 0.08 or lower.

Answer: Thank you. We have now revised the sentence. Please see lines 252-256.

Line 232: students' working status: Please briefly define at some point how you differentiate worker students from non-worker students. Several aspects should be considered in my opinion.
How many hours of work per week are required to be considered as a worker student? Would one hour be enough? How regularly should the student be engaged in working? Should he/she work every week on a regular basis? Or am I also considered as a WS when I was working in the last semester break three months ago? I think these cases should be clearly discussed, if not included in the analyses. In my opinion, regarding the acute demands and stress (at the time point when the measurements took place), it makes a difference if I have to work during the assessment week or if my last job was three months ago.

Answer: Employment status was determined by a single item asking whether the student was regularly employed in the last academic year (yes or no as possible responses). Employed student means both to be a full or a part-time worker. Effectively, the number of work hours can make a difference in stress levels. Due to this caveat in our collecting data, we looked for other studies supporting us in this direction to decide whether to continue in that research line (e.g., Schramer, K.M., Rauti, C.M., Kartolo, A.B., & Kwantes, C.T. (2020), "Examining burnout in employed university students", Journal of Public Mental Health, 19 (1),17-25. https://doi.org/10.1108/JPMH-05-2019-0058). Finally, we added some comments also in the limits section. Please see lines 490-494

Lines 234 to 236: Language; you did not perform measurement invariance or internal consistency. You conducted some analyses to verify these criteria.

Answer: Yes, you’re right. We’ve corrected it.

Line 235: I suggest using the term internal consistency, both here and throughout the manuscript.

Answer: Thank you for your suggestion. We have replaced the term “reliability” with “consistency” throughout the whole manuscript.

Results

Lines 239 and 247: ...shows the fit indizes as well as ΔRMSEA and ΔCFI for...

Answer: We have replaced “fit indices” with “fit indexes” throughout the manuscript.

Lines 239 to 241: Please again briefly describe the factor structure that you modeled.

Answer: We have briefly described the factor structure we modeled through the measurement invariance model. Please see lines 273-285.

Line 244: for instead of between

Answer: Thank you, we revised and corrected this part.

Lines 251 to 252: ...the BAT-C short version worked well in our sample. --> This is too vague and does not transport any specific information.

Answer: Thank you for this important suggestion. Since we reached the strict invariance model, we have now highlighted better than the BAT-C short version can be considered a valid instrument to assess burnout symptoms regardless of worker status differences in university students. Please see lines 294-299.

Lines 252 to 253: In detail, reaching the strict measurement invariance model, ... --> Language, please consider editing.

Answer: Yes, we have revised the editing of this sentence. In detail, we have deleted the sentence “reaching the strict measurement invariance model” because it was redundant.

Table 3: Please indicate at an earlier point of the manuscript that you calculated sum scores for the subscales and not mean scores. Besides, I would recommend to use mean scores for the subscales since these are more intuitive knowing that you used 5-point Likert scales.

Answer: Thank you for your suggestion. We recalculated totals as averages and conducted the analyses again, also in light of the comments made by the Editor. Therefore, we have included a specification concerning the calculation of totals as average scores in the Data analysis section, following your suggestion. Please see lines 267-270.

Table 3: You might want to discuss a potential bias in the ratings of emotional impairment since men might tend more to disguise their emotions towards others and also towards themselves or might feel weak when showing (or indicating) emotional problems.

Answer: Thank you for this valuable comment. We welcomed your suggestion, but we thought it would be more appropriate to include this specification with further elaboration in the discussion section rather than in the results section. We hope that the reported changes will meet your requirements. Thank you.

Table 3: In view of hypothesis 2, I feel it would make sense to indicate the values on the burnout dimensions not only for female and male students and WS and NWS but also specifically for the subgroups stemming from the combination of these two factors.

Answer: Thank you for this important suggestion. We have added in Table 3 the descriptive results (means and standard deviations) for the subgroups (i.e., worker males, no worker males, worker females, no worker females).

Table 3: Please also indicate what ** and Æž2p mean.

Answer: We have added this information in the note below Table 3.

Lines 264 to 265: This should be the first sentence after the table because an at least acceptable internal consistency is an important requirement for the analyses that are to follow. Additionally, I think that the omega of the mental distance subscale is not really good. This should be stated more clearly.

Answer: Thank you for this important suggestion. We moved the internal consistency part immediately after the table, as suggested. In addition, we have specified that the mental distance dimension has an acceptable omega in contrast to the other dimensions, which have excellent scores. Finally, we have included this aspect within the limits of our study.

Lines 270 to 272: Language; consider editing.

Answer: We have revised this part.

Discussion

Lines 283 to 300: From line 283 to line 300 you elaborate on your analysis and results regarding the measurement invariance of the BAT-C. To me this feels like a little bit too much and I suggest to substantially shorten these statements in order to present a brief summary of what you have done, why you have done that and what you found out regarding measurement invariance. The way you have described it so far, it seems like a main result although it rather is the verification of a basic requirement for the following analyses.

Answer: We agree with you and shorten the long introduction to the results comment, reducing the measurement invariance remarks.

Lines 298 to 300: Language

Answer: According to the above concerns, this part is now canceled.

Lines 302 to 305: Please see the comments above concerning the justification of this hypothesis.

Answer: We hope we have disambiguated this point with clearer arguments regarding the expected female students' higher vulnerability to burnout than males. Please see the Introduction paragraph and hypotheses section.

Line 313: In the section below, ... / In the following paragraph, ...

Answer: Ok. Done.

Line 315: In line with...

Answer: Ok. Done.

Line 316: ... are affected by students' gender ... --> This might be correct when speaking about it from a purely statistical point of view. However, you cannot say without a doubt that gender has a direct effect on the BAT dimensions. You may say that female and male university students differ in most of the dimensions.

Answer: Yes, we completely agree with your observation. Now the sentences are more accurate at that point.

Lines 328 and 329: ...higher cognitive and emotional impairment than male university students, ...?

Answer: Yes, corrected.

Lines 332 to 338: I think the relationship between burnout and cognitive impairment/functioning should be put in perspective. The relationship is not really surprising since cognitive impairment represents one of the aspects with which burnout is assessed in this study.

Answer: Ok. We clarified that point. See lines 394-405.

Line 345: ... does not surprise us. --> I recommend to re-phrase this sentence since the reader might ask why you did not present a more specific hypothesis 1 given the fact that you already assumed that there are no differences in MD between females and males.

Answer: We completely agree with you. Thank you. We’ve re-phrased the sentence.

Lines 352 to 359: Here, I do not understand the point you want to make. I do not see the connection between the non-existent differences in MD between females and males and the suggestion of using a person-oriented approach. Please clarify.

Answer: Yes, done.

Lines 360 to 366: I suggest shortening the first part of the paragraph to avoid repetitions.

Answer: Yes, done.

Lines 368 and 369: no repetition needed

Answer: Yes, done.

Lines 372 to 390: This paragraph contains valuable thoughts in my opinion. However, these thoughts do not relate closely enough to the present study and its findings, which makes it hard to follow. I have the feeling that the reader needs to find his/her own way of how to understand what is written in the text. Therefore, I think it would be helpful to make the connection of these considerations to what has been assessed and to what has been found out easier to understand.

Answer: We agree with you. Thank you. We’ve re-phrased the sentence and shortened some pieces.

I recommend to include a paragraph in the discussion section that clearly describes the strengths and limitations of the study.

It would be good to include a paragraph already in the discussion section that outlines some concrete recommendations regarding what could be done to prevent student burnout in the risk group of female university students. Please explain in more detail what could be deduced from the results.

Answer: Yes, we did it in the new conclusions paragraph.

Conclusions

Line 394: student populations

Answer: Yes, done.

Lines 406 to 410: This is already written in the first paragraph of the section.

Answer: Yes, we canceled any repetitions and re-formulated the paragraph.

The conclusion section would benefit from two or three clear messages that describe what the study findings mean for the life of university students.

Answer: Yes, done.

Reviewer 3 Report

This is a well-written manuscript on a very interesting topic. The introduction section sufficiently explains the background of the topic, justification of the problem to be investigated, and a gap in the literature. This section has been organized appropriately. The authors have tried to provide an extensive number of recent research studies as empirical support for the gap and hypothetical relationships. There are some very good studies that the authors have cited. Further, the methodology section is well-written. the sample size has been described appropriately. However, here I would suggest providing in few lines about the population of the study such as who, and where. how and total etc. Likewise, could be better to indicate the type of sample techniques adopted and how the sample was calculated or decided. Moreover, it could be better if the author explain the participants' selection criteria for this study including exclusion and inclusion criteria. Also, could be better if the authors explain in a few lines why the MANOVA test was used in this study. Further results have been provided with sufficient discussion and tabulated data. However, could be better to provide scale reliability values as well. Lastly, I would suggest providing the questionnaire adopted/adapted for this study as an appendix for a better understanding of the results of the study. The authors are suggested to provide limitation /delimitation of this study along with some implications of the findings in one or separate paragraph before the conclusions of the study.     

Author Response

Reviewer 3.

Dear Reviewer,

Thank you very much for your valuable revision. We agree with your concerns and are grateful for your useful suggestions and corrections. We hope you'll find the revised manuscript improved and worthy of publication.

In the following section, you'll find how we revised the manuscript step-by-step.

Sincerely,

The Authors

This is a well-written manuscript on a very interesting topic. The introduction section sufficiently explains the background of the topic, justification of the problem to be investigated, and a gap in the literature. This section has been organized appropriately. The authors have tried to provide an extensive number of recent research studies as empirical support for the gap and hypothetical relationships. There are some very good studies that the authors have cited. Further, the methodology section is well-written. the sample size has been described appropriately.

However, here I would suggest providing in few lines about the population of the study such as who, and where. how and total etc. Likewise, could be better to indicate the type of sample techniques adopted and how the sample was calculated or decided. Moreover, it could be better if the author explain the participants' selection criteria for this study including exclusion and inclusion criteria.

Answer: Thank you for your recommendations. You’ll find more details about participants and recruitment methods in the Participants and Procedure sections. More than inclusion/exclusion criteria, which refer to a controlled sampling method, we described how we obtained the final sample of the current study. Please see lines 194-215.

Also, could be better if the authors explain in a few lines why the MANOVA test was used in this study.

Answer: We have better clarified why the MANOVA test has been employed in the present study. Specifically, although the BAT-C dimensions are distinct, several previous studies highlighted that they are theoretically and empirically correlated (see, for example, Schaufeli et al., 2019). Please see lines 262-270.

Further results have been provided with sufficient discussion and tabulated data.

However, could be better to provide scale reliability values as well.

Answer: Around lines 248-253, you’ll find details answering your requests. If it is not enough, we provide more details.

Lastly, I would suggest providing the questionnaire adopted/adapted for this study as an appendix for a better understanding of the results of the study.

Answer: Thank you for your suggestion. We inserted the BAT-C short version in the Appendix section.

The authors are suggested to provide limitation /delimitation of this study along with some implications of the findings in one or separate paragraph before the conclusions of the study.

Answer: Yes, we agree with you. Done.  

Reviewer 4 Report

I feel that the biggest issue with this paper is in what the survey purports to measure. I think there are too many variables on what can be driving burnout to be relegated to what's simplistically asked using BAT tool. Variables pertaining to what are the stressors, how long experiencing, what and how long remedial/treatments sought, all of these are important questions to ask. The respondents' demographic background also needs to be analyzed further.

I think all of these are lacking in this paper.

Author Response

Dear Reviewer,

Thank you very much for your valuable revision. We agree with your concerns and are grateful for your useful suggestions and corrections. We hope you'll find the revised manuscript improved and worthy of publication.

In the following section, you'll find how we revised the manuscript step-by-step.

Sincerely,

The Authors

Reviewer 4.

I feel that the biggest issue with this paper is in what the survey purports to measure.

I think there are too many variables on what can be driving burnout to be relegated to what's simplistically asked using BAT tool. Variables pertaining to what are the stressors, how long experiencing, what and how long remedial/treatments sought, all of these are important questions to ask. The respondents' demographic background also needs to be analyzed further.

I think all of these are lacking in this paper.

Answer: Thank you for your suggestions. We used the BAT-C short form in the current study as a first measurement invariance analysis as a starting point for further instrument implementations. We don’t think, like your opinion, that it would be enough to obtain a clinical diagnosis of students' burnout syndrome. In this regard, we’ve further explained that point in the limits and future studies paragraph.

Moreover, we added more details regarding participants’ characteristics. Please see the red lines in proper paragraphs.

Reviewer 5 Report

Point one: Research structure

I recommend restructuring this manuscript according to the following headings.

Abstract

1. Introduction

2. Literature Review

2.1. Previous Research

2.2. Theoretical Foundation

2.3 Proposed Model and Hypotheses Formulation

3. Materials and Methods

3.1 Research context

3.2 Sample and data collection

4. Results and discussion

5. Theoretical and Practical Implications

6. Limitations and future research recommendations

7. Conclusion

Point two: Abstract part

The abstract need to be restructured according to the followings points;

·         The objective of the study

·         Design/methodology/approach

·         Findings

·         Practical implications

·         Originality/value.

The authors clarify the objective of the study while the design, method, and approach need to improve. The rest of the three points did not cover.

Point three: Title

·         The title is not well formulated and the title does not fully reflect the topic and nature of the study, but only partially

·         I suggest the following title: factors influencing  Students' Burnout During Covid-19 Pandemic among Universities: The Role of Gender and Worker Status 

Point four: Introduction part

·         The introduction is understandable, clear and closely related to the research problem. Also, the research problem is scientifically formulated correctly.

·         Research objectives are not specific and unclear formulated. In addition, the importance of research is unclear.

·         The introduction is not directly related to the research title. Also, The main variables in the research are not addressed in the introduction. Authors must add them.

 Point five: The Students' Burnout at University in Italia

·         I believe this part is necessary. Many ideas have been repeated in the introduction.  I recommend concise paragraphs to one or two paragraphs about what is happening in Italia and adding it to the introduction part. Also, by adding a section in the methodology part under the name "Research context " you can address all important points about the research context. 

Point six: Theoretical framework & Research Hypotheses parts

·         This part is not clear enough,  I recommend restructuring and creating it according to the following

2. Literature Review

2.1. Previous Research

2.2. Theoretical Foundation

2.3 Proposed Model and Hypotheses Formulation

 The author should represent the research framework showing their hypothesis, and the results should be validated in the model.

Point seven: methodology part

·         The methodology part is a bit confusing, the researchers did not specify whether this study was an Inductive or Deductive Research Approach. Many studies conducting begin with an inductive study (developing a theory). The inductive study is followed up with deductive research to confirm or invalidate the conclusion. The conclusion (theory) of the inductive study is also used as a starting point for the deductive study. Authors must provide a scientific justification for the employed approach with references to show The validity of the approach used in the study. 

·         The authors did not clarify the sampling technique (the study includes a sample of 494 university students, how they determined the sample frame of the study -how they choose the included students in the study- and based on what, and how they calculated the Sample size ).

·          The methodology part needs more clarifications.  

Point eight: Results and discussions part

The research presented the results in an accurate and organized manner, and sound scientific methods were employed in interpreting them. Additionally, the results were compared with those of previous research that supports them.

Point nine: Conclusion part

I recommend revising the structure of the last section: restating the topic, restating research objectives, summarizing the main points of the research, discussing the significance of these points, and presenting future research directions. In addition, an enclosing (positive) paragraph comprising a “great conclusion” that makes the readers think would be appropriate. I advise you to choose a more appropriate and thought-provoking way to end your paper. Comparing your ideas to a universal concept may help readers relate and may provide a better sense of closure (since your readers will find your ideas more compelling if they can relate to them).

Authors should tell theoretical and managerial implications of their study

Author Response

Dear Reviewer,

Thank you very much for your valuable revision. We agree with your concerns and are grateful for your useful suggestions and corrections. We hope you'll find the revised manuscript improved and worthy of publication.

In the following section, you'll find how we revised the manuscript step-by-step.

Sincerely,

The Authors

Reviewer 5.

Point one: Research structure

I recommend restructuring this manuscript according to the following headings.

Abstract

  1. Introduction
  2. Literature Review

2.1. Previous Research

2.2. Theoretical Foundation

2.3 Proposed Model and Hypotheses Formulation

  1. Materials and Methods

3.1 Research context

3.2 Sample and data collection

  1. Results and discussion
  2. Theoretical and Practical Implications
  3. Limitations and future research recommendations
  4. Conclusion

Answer: Thank you for your advice. We have changed some paragraphs’ structures according to the MDPI’s guidelines as well as the kind of study we have submitted. You may see all changes in the red lines throughout the whole document.

Point two: Abstract part

The abstract need to be restructured according to the followings points;

  • The objective of the study
  • Design/methodology/approach
  • Findings
  • Practical implications
  • Originality/value.

The authors clarify the objective of the study while the design, method, and approach need to improve. The rest of the three points did not cover.

Answer: Thank you. We have revised the abstract, as shown by the red lines. We agree with your concerns, but please consider that we must save the limits of the abstract’s words.

Point three: Title

The title is not well formulated and the title does not fully reflect the topic and nature of the study, but only partially.  I suggest the following title: factors influencing  Students' Burnout During Covid-19 Pandemic among Universities: The Role of Gender and Worker Status 

Answer: Thank you for your suggestion. However, we think that by adding the keyword ‘Covid-19’, we alert the reader to a variable that we have not exactly investigated and for which we don’t have any comparative sample. It was the first time we used BAT-C in a sample of Italian university students.

Point four: Introduction part

  • The introduction is understandable, clear and closely related to the research problem. Also, the research problem is scientifically formulated correctly.
  • Research objectives are not specific and unclear formulated. In addition, the importance of research is unclear.
  • The introduction is not directly related to the research title. Also, The main variables in the research are not addressed in the introduction. Authors must add them.

Answer: Thank you for your suggestions. We partially agree with you by considering that it is important to keep meaning and research focus without any redundant arguments. We added some more details, as shown in the red lines.

Point five: The Students' Burnout at University in Italia

I believe this part is necessary. Many ideas have been repeated in the introduction.  I recommend concise paragraphs to one or two paragraphs about what is happening in Italia and adding it to the introduction part. Also, by adding a section in the methodology part under the name "Research context " you can address all important points about the research context. 

Answer: we added more details concerning your requests. See participants and procedure sections.

Point six: Theoretical framework & Research Hypotheses parts

This part is not clear enough, I recommend restructuring and creating it according to the following

  1. Literature Review

2.1. Previous Research

2.2. Theoretical Foundation

2.3 Proposed Model and Hypotheses Formulation

The author should represent the research framework showing their hypothesis, and the results should be validated in the model.

Answer: Thank you for your advice. We have changed some paragraphs’ structures according to the MDPI’s guidelines as well as the kind of study we have submitted. You may see all changes in the red lines throughout the whole document.

Point seven: methodology part

The methodology part is a bit confusing, the researchers did not specify whether this study was an Inductive or Deductive Research Approach. Many studies conducting begin with an inductive study (developing a theory). The inductive study is followed up with deductive research to confirm or invalidate the conclusion. The conclusion (theory) of the inductive study is also used as a starting point for the deductive study. Authors must provide a scientific justification for the employed approach with references to show The validity of the approach used in the study. 

Answer: the current study is a cross-sectional study where a new tool has been adopted for the first time for a university sample. With this in mind, we used a methodological approach based on reliable literature. Below you find the paragraph we have introduced in the discussion section before another reviewer’s recommendations asking for a shorter description of that part.

“According to Zumbo and Koh [80], one essential criterion to consider an instrument a valid measure is its usability across multiple socio-demographic groups (in the current study, students’ gender and worker status). Indeed, a good measure should maintain the exact relationship between each item score and the respective factor score in the CFA model across different groups. Considering our findings, we assumed BAT’s invariance across population comparisons. Thus, starting from the above comforting results, we verified our hypotheses concerning the effect of gender and worker status on BAT-C levels”.

The authors did not clarify the sampling technique (the study includes a sample of 494 university students, how they determined the sample frame of the study -how they choose the included students in the study- and based on what, and how they calculated the Sample size ).

Answer: Thank you for your recommendations. We collected data by a snowball sampling method. Because our hypotheses focused on gender differences and worker status, we randomly selected subjects according to two balancing criteria: gender and working status. We added some more details.

The methodology part needs more clarifications. 

Answer: We’d be glad to add more clarifications point. Could you give us more details in this regard?

Point eight: Results and discussions part

The research presented the results in an accurate and organized manner, and sound scientific methods were employed in interpreting them. Additionally, the results were compared with those of previous research that supports them.

Answer: Thank you for that.

Point nine: Conclusion part

I recommend revising the structure of the last section: restating the topic, restating research objectives, summarizing the main points of the research, discussing the significance of these points, and presenting future research directions. In addition, an enclosing (positive) paragraph comprising a “great conclusion” that makes the readers think would be appropriate. I advise you to choose a more appropriate and thought-provoking way to end your paper. Comparing your ideas to a universal concept may help readers relate and may provide a better sense of closure (since your readers will find your ideas more compelling if they can relate to them).

Authors should tell theoretical and managerial implications of their study

Answer: Thank you for your recommendations. We now added more argumentations and, hopefully, open mind perspective for further research and practical implications.

Round 2

Reviewer 2 Report

Thank you for submitting the revised version of your mansucript, which I very much enjoyed reading. In my opinion, the manusript has substantially improved. However, there are still some points that I would suggest to address.

Introduction

Line 44: the students

Line 110: feel more exhausted

Line 115: compared to

Line 121: delete “that”

Line 177: delete “highlights”

Lines 182-183: Concerning the gender effect, we expected hypothesized that female university students were more burned out than male students (H1). à In my opinion you make it really clear that the previous findings regarding gender differences are highly diverse. With regard to that, I still think that H1 is a little bit too strong. I wonder why you do not present it as an undirected hypotheses, which would simply assume differences between gender groups. This aspect also applies to your statements in the discussion section, lines 354 to 356.

Materials and Methods

Line 228: previous instead of last, to avoid any misunderstandings

Line 248: …when a further constraint is added.

Results

We have briefly described the factor structure we modeled through the measurement invariance model. Please see lines 273-285. à You did not describe the factor structure in these lines, you described the stepwise procedure that you followed in investigating measurement invariance (which I like very much). With my original comment I intended to ask you whether you would like to repeat the assumed 4x3 items factor structure with which you try to assess the four dimensions.

Discussion

Line 400: postulates

Original comment from round 1: Lines 332 to 338: I think the relationship between burnout and cognitive impairment/functioning should be put in perspective. The relationship is not really surprising since cognitive impairment represents one of the aspects with which burnout is assessed in this study.

Answer: Ok. We clarified that point. See lines 394-405. à In lines 394-405 of the revised manuscript you write about considerations regarding emotional impairment. In my original comment, I referred to cognitive impairment. In my opinion, it would be good to mention that the relationship between burnout (they way you assessed it) and cognitive impairment is not surprising since the cognitive impairment is one subscale of the BAT-C.

Line 429: which instead of that

Line 448: current study

Line 501: in a previous

Lines 511-512: I do not understand what you want to say with “even not sufficient when worker status has been analyzed”. Please edit.

General

I highly recommend to thoroughly go through the manuscript again and eliminate the slips as well as the minor linguistic mistakes.

Author Response

Dear Reviewer,

Once again, thank you for your useful and accurate revision of our paper.

We have reviewed the manuscript taking into account all your corrections and suggestions.

We hope now the paper is more accurate. All changes are in blue lines, and our replies to your comments are interactively put in the following text.

Sincerely,

The Authors

Rev 2 round II

Comments and Suggestions for Authors

Thank you for submitting the revised version of your manuscript, which I very much enjoyed reading. In my opinion, the manuscript has substantially improved. However, there are still some points that I would suggest to address.

Introduction

Line 44: the students: done

Line 110: feel more exhausted done

Line 115: compared to done

Line 121: delete “that” done

Line 177: delete “highlights” done

Lines 182-183: Concerning the gender effect, we expected hypothesized that female university students were more burned out than male students (H1). In my opinion you make it really clear that the previous findings regarding gender differences are highly diverse. With regard to that, I still think that H1 is a little bit too strong. I wonder why you do not present it as an undirected hypotheses, which would simply assume differences between gender groups. This aspect also applies to your statements in the discussion section, lines 354 to 356.

Answer (1): we accepted your suggestions, and now we propose H1 in a way more coherent with the literature. In this regard, several changes have been introduced: see the abstract section, hypothesis paragraph, discussion, and conclusion. Please see the new blues lines and relative canceled sentences. 

Materials and Methods

Line 228: previous instead of last, to avoid any misunderstandings done

Line 248: …when a further constraint is added. done

Results

We have briefly described the factor structure we modeled through the measurement invariance model. Please see lines 273-285. You did not describe the factor structure in these lines, you described the stepwise procedure that you followed in investigating measurement invariance (which I like very much). With my original comment I intended to ask you whether you would like to repeat the assumed 4x3 items factor structure with which you try to assess the four dimensions.

Answer (2): In our previous study, we have already verified and confirmed the factorial structure, reliability, and validity of the BAT-C for students (Romano et al., 2022). In the current study, the confirmatory factor analysis of the BAT-C wasn’t the main aim, but it is just one step of the measurement invariance analysis (Little, 2013; Cheung et al., 2009). While to meet your concerns, we now have described in detail the results concerning the main hypotheses. Is it clear enough in the text?

Discussion

Line 400: postulates done

Original comment from round 1: Lines 332 to 338: I think the relationship between burnout and cognitive impairment/functioning should be put in perspective. The relationship is not really surprising since cognitive impairment represents one of the aspects with which burnout is assessed in this study.

Answer: Ok. We clarified that point. See lines 394-405. à In lines 394-405 of the revised manuscript you write about considerations regarding emotional impairment. In my original comment, I referred to cognitive impairment. In my opinion, it would be good to mention that the relationship between burnout (they way you assessed it) and cognitive impairment is not surprising since the cognitive impairment is one subscale of the BAT-C.

Answer (3): ok. Now we added some lines in this regard. Please, see blues lines 404-405.

Line 429: which instead of that done

Line 448: current study done

Line 501: in a previous done

Lines 511-512: I do not understand what you want to say with “even not sufficient when worker status has been analyzed”. Please edit. done

General

I highly recommend to thoroughly go through the manuscript again and eliminate the slips as well as the minor linguistic mistakes. done

Reviewer 4 Report

For the current revised version of the paper, I think the fundamental flaw is still there, i.e., the paper is basically investigating how gender and work status may affect one's propensity to become burn out. Overall, I think the research objective of the paper is too simplistic and as a reader, I really did not learn anything new without the paper concurrently testing other variables. As I mentioned before in my prior reviewer comments, the important omitted variables cannot be just left as limitations, or else the findings will not illuminate anything new that one would not have known before reading the paper.

In my opinion, to make the paper into a publishable state, author should consider rerunning a new survey again with other important and control variables that I have mentioned in my prior review.

Author Response

Dear Reviewer,

Thank you for your comments and suggestions.

We have reviewed the manuscript considering your concerns regarding other confounding student variables (i.e., the type of study and enrollment year). After a chi-square test that highlighted a significant association between gender and academic curricula, we controlled for the latter, entering it as a covariate in the MANCOVA.

Please, read all new analyses and comments from lines 318 to 347.

We hope now the paper is more adherent to your requests.

Sincerely,

The Authors

Rev 4 round II

For the current revised version of the paper, I think the fundamental flaw is still there, i.e., the paper is basically investigating how gender and work status may affect one's propensity to become burn out. Overall, I think the research objective of the paper is too simplistic and as a reader, I really did not learn anything new without the paper concurrently testing other variables. As I mentioned before in my prior reviewer comments, the important omitted variables cannot be just left as limitations, or else the findings will not illuminate anything new that one would not have known before reading the paper.

In my opinion, to make the paper into a publishable state, author should consider rerunning a new survey again with other important and control variables that I have mentioned in my prior review.

Reviewer 5 Report

Thank you for your revision to my comments. I think the authors successfully revised most of my comments. However, there is still an unsolved issue regarding the methodology partI still do not see the clearly presented.  In the methods section, even though you have success to explain the ethical issue, however, you must properly describe the study population, the study procedure, and the analysis strategy. All these sections are missing in the methods section. Inclusion and exclusion criteria must be reported. Any excluded subject? You should precisely report how many subjects refused to participate. How did you enroll participants? The authors did not directly answer this question and they asked for more details. 

I suggest additional reading that will surely help increase the study’s general impact, taking into consideration the international perspective. References (to be consulted and not necessarily to be cited):

1- Alsyouf, A., Masa’deh, R. E., Albugami, M., Al-Bsheish, M., Lutfi, A., & Alsubahi, N. (2021). Risk of Fear and Anxiety in Utilising Health App Surveillance Due to COVID-19: Gender Differences Analysis. Risks9(10), 179.

2- Althunibat, A., Altarawneh, F., Dawood, R., & Almaiah, M. A. (2022). Propose a New Quality Model for M-Learning Application in Light of COVID-19. Mobile Information Systems2022.

3- Almaiah, M. A., Hajjej, F., Lutfi, A., Al-Khasawneh, A., Alkhdour, T., Almomani, O., & Shehab, R. (2022). A Conceptual Framework for Determining Quality Requirements for Mobile Learning Applications Using Delphi Method. Electronics11(5), 788.

Best regards

Author Response

Dear Reviewer,

Thank you for your comments and suggestions. We have reviewed the manuscript considering your concerns regarding more details of samples’ characteristics (Table 3, comments, and notes in blue lines). Moreover, as you can see in the latest reviewed manuscript, all information regarding the method of data collection and analysis strategy was already specified.

Finally, it is hard for us to find a relevant connection between the suggested references and the focus of our study. We hope you may understand our difficulties.

We also hope that now the paper is more adherent to your requests.

Sincerely,

The Authors